# Pretrained Language Models are Symbolic Mathematics Solvers too!

## Abstract

Solving symbolic mathematics has always been of in the arena of human ingenuity that needs compositional reasoning and recurrence. However, recent studies have shown that large scale language models such as transformers are universal and surprisingly can be trained as a sequence-to-sequence task to solve complex mathematical equations. These large transformer models need humongous amounts of training data to generalize to unseen symbolic mathematics problems. In this paper, we present a sample efficient way of solving the symbolic tasks by first pretraining the transformer model with language translation and then fine-tuning the pretrained transformer model to solve the downstream task of symbolic mathematics. We achieve comparable accuracy on the integration task with our pretrained model while using around 1.5 orders of magnitude less number of training samples with respect to the state-of-the-art deep learning for symbolic mathematics. The test accuracy on differential equation tasks is considerably lower comparing with integration as they need higher order recursions that are not present in language translations. We pretrain our model with different pairs of language translations. Our results show language bias in solving symbolic mathematics tasks. Finally, we study the robustness of the fine-tuned model on symbolic math tasks against distribution shift, and our approach generalizes better in distribution shift scenarios for the function integration.

## 1 Introduction

Deep learning is a ubiquitous choice in solving statistical pattern recognition problems of regression and classification. With a large training data set and compute power, they have proven to be very effective and achieve state-of-the-art performance in a wide range of tasks in natural language processing, computer vision, speech recognition, sentiment analysis, etc (Lu et al., 2021). Though deep learning triumphs in the statistical domain (Bengio et al., 2003), there is an active interest in extending deep networks in symbolic computation (Lample & Charton, 2019; Davis, 2019; Allamanis et al., 2017; Zaremba et al., 2014; Loos et al., 2017). There are mainly two motivations for this: (i) performing symbolic mathematical tasks, such as symbolic integration and solving differential equations, in deep net architectures, and (ii) applying neural networks in the domain automated theorem proving, computer algebra systems, and natural language understanding (NLU) that requires a symbolic knowledge system. The key capability of symbolic computation is that symbols maintain their identity as they do multiple roles while deep neural networks exploit shared representation and composition.

This paper uses a pretrained language model to solve symbolic mathematics tasks, particularly symbolic integration and differential equations. We show our pretrained transformer architecture on language translation is expressive enough to solve large class symbolic mathematics such as function integration and differential equations, which have traditionally been approached using logic and exhaustive search. Moreover, our pretrained model is *sample efficient* and compute efficient–i.e., it requires fewer epochs to converge to good accuracy. The first major work of solving symbolic mathematics with transformer architecture is by Lample & Charton (2019). They use the transformer model that is mainly used for NLP

tasks to solve the symbolic computation. They first re-frame the mathematical equations as text sequences and then solving those equations as a sequence-to-sequence translation. Their transformer model catches pattern in the mathematical expressions, e.g., the expressions of the form $\sin^{-1}(x)$ will have its primitive as $\frac{1}{\sqrt{1-x^2}}$. We extend the work of Lample & Charton (2019) and train their symbolic math dataset by fine-tuning pretrained translation models to solve the downstream task of symbolic mathematics. The pretrained language model will transfer the syntactic and semantic structure of the present in the language, mathematical expressions represented as trees. The inherent limitation between the many-to-one map between mathematical expression and tree encoding is partially regularized by the pre-training with the language translation. For example, the same mathematical expressions $7 + 3 \times (5 + 2)$ and $3 \times (5 + 2) + 7$ are represented are encoded as different trees. We regularize (penalize) this freedom of expression of encoding a mathematical expression by multiple trees by pretraining our transformer model with language translation. The sentence in a language has an order as specified by the famous quote by J. R. Firt "You shall know a word by the company it keeps.". Unlike language, where the meaning of a word is given by its neighbors, the value of a mathematical sub-expression (mathematical word) is not influenced by its neighboring expressions. In their training data set generation for function integration, mathematical expressions $F$ and $G$ are generated and the corresponding derivatives $f$ and $g$ are computed. The training data set are the tuples $(f, F)$, $(g, G)$ and a new integration function data set $Fg$ is generated (assuming $(fG, \int fG)$ is in the training set) through IBP (Integration By Parts) [1] method as:

$$\int Fg = FG - \int fG.$$

Their vanilla transformer model during training learns to build the correlation between $\int Fg$ and $fG$ for solving symbolic mathematics. We differ from their model by (i) forcing our transformer model to develop conditional probability between randomly generated functions $P_\Theta(f|G)$ and $P_\Theta(g|F)$ as follows:

$$P(fG) = P(f|G)P(G)$$
$$P(Fg) = P(g|F)P(F)$$

where $P_\Theta$ is our pretrained transformer model and $\Theta$ is the learned parameter (weights and biases). By re-framing the problem to a conditional probability model, we bypass the distributions of randomly generated functions $P(F)$ and $P(G)$. Our method also shows marginal robustness to different types of data set generation method, as shown in table 4. (ii) Our model's predictions improve even when there is a difference of length between input and output sequence. This is because of the phenomena of heavy-tailed distribution, where the model can generate rare small or large output expressions (Sornette, 2006; Martin & Mahoney, 2018). Our model is less sensitive with large difference of length between input and output mathematical expressions (i.e., the problem and the solution sequence.) as explained in Section 3.

The paper is organized as follows: In Section 2 we discuss the prediction of our pretrained transformer model in the language conditional probability and optimization, Section 3 discusses our proposed of heavy-tailed self-regularization under the mild condition of our pretraining, Section 4 discusses experimental setting and methodology, architecture, datasets, and the evaluation metric, and Section 5 poses the following research questions and answers them:

1. Does this pretrained model help us to use less data for fine-tuning?

2. Does the result of this fine-tuning depend on the languages used for pretraining?

3. How robust this fine-tuned model is with respect to distribution shift of test data in comparison with fine-tuning data?

Section 6, discusses literature review, and finally, Section 7 concludes the paper.

---

[1] More details about the datasets are explained in section 4.2.

## 2 Problem Formulation

Mathematical expressions can be depicted as binary-unary trees, with operators in the form of internal nodes, operands in the form of children, and numbers, constants, and variables in the form of leaves (Lample & Charton, 2019). These trees can be transformed into a sequence of math letters by traversing them in some specific order. In this paper, a tree of symbolic mathematical expressions is scanned by the prefix traversal to produce a sequence corresponding to the mathematical expression. We formulate our symbolic mathematics as a Seq2Seq translation problem with a large scale pretrained mBART (and Marian-MT) transformer. The pretrained transformer is retrained with random expressions data set for the case function integration and differential equation.

The training dataset for both tasks is a tuple of mathematical expressions in the format of $(f_{\text{problem}}, f_{\text{solution}})$. Our pretrained transformer model $P_\Theta$ solves the symbolic mathematics task by minimizing the prediction loss $e$ as follows:

$$\underset{\Theta}{\text{minimize}} \frac{1}{n} \sum_{i=1}^{n} e(P_\Theta(f_{\text{problem}}), f_{\text{solution}}) \tag{1}$$

where $\Theta$ is the learned parameter and $n$ is the number of samples.

## 3 Theory

Pretraining of the mBART transformer $P_\Theta$ is done by Seq2Seq translation between the source language English and target language of Romanian. The model parameters $\Theta$ as expressed in Equation 1 are learned with gradient descent during the translation task. Fine-tuning the model on symbolic data set allows the model to transfer the knowledge of language translation. Our model inputs the mathematical expressions and predicts the output mathematical expressions of the shortest length. The model searches its big hypothesis space and finds the optimum hypothesis that outputs the shortest mathematical sequence. Searching the big hypothesis space incurs huge optimization cost and also the optimization surface is non-convex. Our model generalizes using the phenomena of self-regularization, which is a complex interpolation process between two parameter spaces. The parameter space $\Theta$ of our transformer model lie in a very high-dimensional space and is poorly understood. We model our parameter space as a Normed space. The learned parameter space $\Theta_{translation} = (R^d, ||.||_{\Theta_{translation}})$ is a $d$-dimensional normed space where $d$ is very large. The learned parameter space comes after the training from English to Romanian translation. Fine-tuning the model with mathematical expressions is a perturbation of the parameter space $\Theta_{translation}$ to a new parameter space $\Theta_{fine-tuning}$. The fine-tuning normed space can similarly be defined as $\Theta_{fine-tuning} = (R^d, ||.||_{\Theta_{fine-tuning}})$. Our transformer model during the fine-tuning does a complex interpolation between normed spaces from $\Theta_{translation}$ to $\Theta_{fine-tuning}$ with a bounded variance (Andoni et al., 2018a;b).

During the fine-tuning of our model with a new test sample $Q \subseteq \Theta_{fine-tuning}$ of symbolic expression, our model does an approximate nearest neighbor approximation search and find closet point in $P \subseteq \Theta_{translation}$ in the translation metric space. For given $c \geq 1$ and $r > 0$ our transformer model finds point $P$ such that distance $d_X(q, p) \leq r$ with $c-$Approximate Near Neighbor Search (c-ANN). The large size of the transformer parameter space, it is randomly partitioned into translation space $\Theta_{translation}$ and fine-tuning space $\Theta_{fine-tuning}$. Each of the partition of translation space and fine-tuning space is prepossessed as a data structure of $Cell_{translation}$ and $Cell_{fine-tuning}$. The $Cell_{translation}$'s are then represented as a binary tree as shown in1. Given a query data structure $Cell_{fine-tuning}$ during the symbolic mathematics tasks, our transformer model searches its $Cell_{fine-tuning}$ binary tree to find the approximate match. The normed space restrictions of $\Theta_{translation}$ and $\Theta_{fine-tuning}$ guarantees the $c$-Approximate Near Neighbor Search (c-ANN) in polynomial time. The self-regularization process comes from the underlying high-dimensional geometric structure of $\Theta_{translation}$ and $\Theta_{fine-tuning}$.

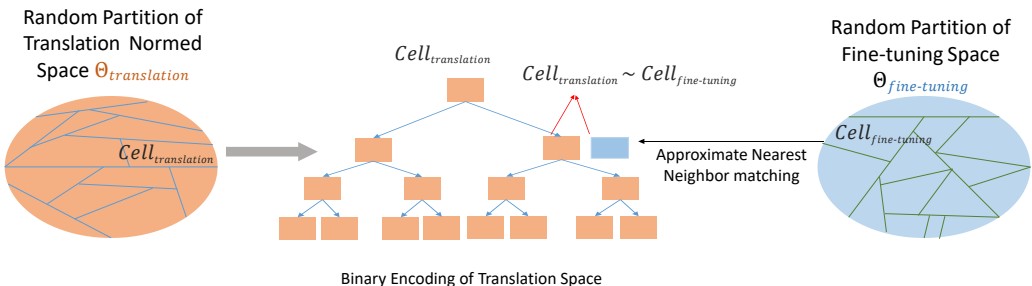

Figure 1: Interpolation between translation space and fine-tuning space.

## 4 EXPERIMENTAL SETTING

We evaluate a diverse set of symbolic mathematical data sets as introduced in Lample & Charton (2019). The tasks studied in these datasets include symbolic integration and solving differential equations of order one and two. Mainly, we are interested in whether pretrained language models are genetically capable of solving these tasks with fewer data. Moreover, whether the language that they have been pretrained on impacts their result after transfer learning. In section 5, we will do this empirical study by asking structured research questions.

### 4.1 ARCHITECTURE

We use the Marian model (Junczys-Dowmunt et al., 2018) and the mBART model (Liu et al., 2020), pre-trained on different translation tasks by the NLP group at the University of Helsinki and Facebook, using the Marian model and the mBART model of the famous NLP framework, Hugging-Face (Wolf et al., 2019).

Both models follow the famous transformer architecture introduced in Vaswani et al. (2017). The Hugging-Face mBART model has an embedding size of 1024, with 16 attention heads and 12 layers. The Marian-MT model we use (only) in section 5.2, has an embedding size of 512, with 8 attention heads and 6 layers. The Marian Model and the mBART model have approximately 74 and 610 a million parameters. The Parameter counts may vary depending on the vocab size of the language they have been pretrained on. We also train the model used in Lample & Charton (2019) with the same parameters as the mBart model (i.e., with an embedding size of 1024, 12 layers and 16 attention heads.).

### 4.2 DATASETS

Thanks to Lample & Charton (2019), there is a good dataset resource for Symbolic Mathematics available publicly. In all the experiments in this paper, we use the same datasets as Lample & Charton (2019), or generate new datasets using the same generation methods.

For the mathematical integration task, there are three generation methods. Forward (FWD), Backward (BWD), and Integration by Parts (IBP). The forward approach, generates random functions and calculates their integrals with an external symbolic mathematical framework. The backward approach, on the other hand, generates a random function and then computes its derivative and add the pair to the dataset with a backward manner. Both backward and forward approaches have some issues. The forward approach is only capable of creating samples that can only be integrated by a mathematical framework and also the samples generated by this approach have short problems with long solutions. The backward approach normally generates samples that the integral is shorter than the equation itself. In contrast to the other two methods, the IBP approach uses the integration by parts formula to generate

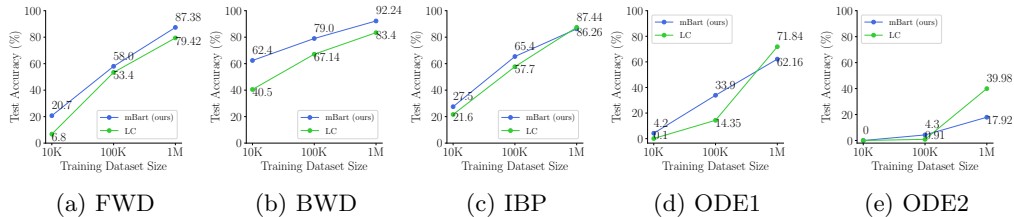

Figure 2: The accuracy of our mBART language model and the LC model when trained on different training sample sizes. Panels (a), (b), and (c) are for the integration task.

Table 1: Accuracy of our models (in percentage (%)) and the Lample & Charton (2019)'s model on integration and differential equation solving. The number of training samples used to train the models in all tasks is 1 million. Results are tested on test data sets of size 5000 samples.

|  | Integration (FWD) | Integration (BWD) | Integration (IBP) | ODE 1 | ODE 2 |
|---|---|---|---|---|---|
| Our Model | 87.4 | 92.2 | 86.2 | 62.2 | 17.9 |
| LC's Model | 79.4 | 83.4 | 87.4 | 71.8 | 39.9 |

samples that do not need an external computer algebra framework, but in terms of the equation lengths, it is similar to the FWD approach (generates short problems and long solutions.) (Lample & Charton, 2019). The datasets for the first order differential equations are referred as ODE1 and the second order differential equations are referred as ODE2. Detailed information about datasets can be found at Lample & Charton (2019).

### 4.3 Metric

In all of our experiments, we report the Accuracy, which is defined as follows:

Accuracy: As discussed in the section 4.3 of Lample & Charton (2019), we can easily calculate the accuracy of our predictions by comparing the generated equation and the reference equation. The generated equation by the models might not be in the same format as the reference equation; therefore, we simplify the difference between the predicted and the reference equation to check whether it is 0 or not. It is also necessary to mention that all of the results in section 5 are reported with the evaluations of beam size 1.

## 5 Experimental Evaluation

In this section, we examine the results showing transfer from language translation to solving symbolic math equations and attempt to understand better why this happens and which factors enable this transfer. The following subsections include our research questions, how we design experiments to answer them, the discussions of the results, and their implications. Note that we refer to Lample & Charton (2019)'s model results with the keyword **LC** in our tables and visualizations.

We train our models with the Adam optimizer (Kingma & Ba, 2015), with a learning rate of $10^{-4}$. We run all of our experiments with the mBART and the Marian-MT model only for 15 epochs, while we train the LC model as long as the model converges (which usually takes around 100 epochs.). [2]

### 5.1 Does this pretrained model help us to use less data for training?

As studied in Lample & Charton (2019), to train transformer architecture on the symbolic math data, we need a vast amount of training data for each task to achieve the highest

---

[2]The experiments with the mBART model were performed on a machine equipped with one RTX A6000 NVIDIA GPU and 48GB memory. The experiments with the Marian-MT model were performed on a machine equipped with one NVIDIA Tesla V100 GPU and 512GB memory.

accuracies (in the order of 40 million to 80 million training samples for each task.). We investigate if fine-tuning the pretrained models on language translation tasks on the symbolic math data can help us use considerably fewer data in the fine-tuning stage.

In this section, we will use the pretrained mBART (Liu et al., 2020) model for the English to Romanian translation task [3], and fine-tune it on our math data (see section 4.2). We report the accuracy of our models on the integration and differential equation solving in table 1. In this table, we use the same training dataset for both our mBART model and the LC model. We train our mBART model only for 15 epochs for all 5 tasks (FWD, BWD, IBP, ODE1, and ODE2), but we continue the training of the LC model until convergence (which takes around 100 epochs for each task.). We can see in the table 1 that our model outperformed in the integration task, with a considerable gap from the LC model. But it cannot properly perform on the differential equation task, especially the second-order differential equations.

We extend this exploration by running the same experiment for different orders of magnitude of training data (i.e., 10K, 100K, and 1M). We report the test accuracy (see section 4.3) of each experiment for both models (mBART and LC) in figure 2. Our model has higher accuracy in comparison to LC in all tasks and with different training sample sizes, except that in the differential equations the accuracy growth of our model suddenly gets lower than the LC model when using the 1 million samples for training.

We achieve comparable accuracy on the integration task with our pretrained model while using around 1.5 orders of magnitude less number of training samples than the state-of-the-art model in Lample & Charton (2019) (i.e, we use 1 million training samples against the 40-80 million samples that Lample & Charton (2019) used for training their model.). As we have discussed previously in the section 3, the mBART language model has already been pretrained by the language translation. During this pretraining, our mBART model searches for that hypothesis that outputs the shortest translated sequence (the shortest Romanian sequence for a given input of English sequence). During the fine-tuning, it uses the same hypothesis learned previously to search for mathematical expressions that has minimum length. Also, because our mBART language model is very large, it is doing an internal look-up and search for the solutions depth-wise in the mathematical expression tree. The model is thus effectively searching greedily than the LC model.

Note that the accuracies reported for the LC model in table 1, as well as in tables 3 and 4 are by training this model also with 1 million training samples (the high accuracies (over 95%) reported in Lample & Charton (2019) are achieved by sample sizes of range $20 - 40$ million training samples).

## 5.2 Are the results of such fine-tuning, language dependent?

We investigate whether different languages used to train our pretrained models impact the results of this transfer learning. We wish to see whether the quality of the results in section 5.1 might have been dependent on the specific source-target language in our language model, i.e., the learned representations. In other words, the specific language could have been a confounder. Therefore, to remove this confound, we fine-tune our symbolic math data on 9 different pretrained language translation tasks containing various source-target languages.

To be able to perform more experiments on multiple languages (due to the computational costs), we fix our training sample size to 100K samples per task, and we use the pretrained Marian-MT model of Hugging-Face (Wolf et al., 2019) which has already been pretrained on many language translation tasks, and is available online [4]. Since the accuracy of the models based on what we saw in section 5.1 are consistent, we only report the accuracies for the 100K sample dataset. Accuracies will not be optimal, but they are sufficient to answer our question. We test all the experiments on test datasets of size 1000. The results are shown in table 2. As we can see in this table, for each task, a different pretrained language has the highest accuracy (indicated in bold case.). For example, in the FWD task the French to

---

[3]The pretrained mBART model is available at https://huggingface.co/facebook/mBART-large-en-ro.

[4]The pretrained Marian-MT models are available at https://huggingface.co/Helsinki-NLP.

Table 2: Evaluation of accuracy of our Marian-MT model (in percentage (%)) on the integration and differential equation solving for different pretrained languages. The highest accuracy is indicated by bold case in each column (task). We see that the language has no specific impact on the results of this fine-tuning.

| Language | Integration (FWD) | Integration (BWD) | Integration (IBP) | ODE 1 | ODE 2 |
|---|---|---|---|---|---|
| English - Romanian | 38.8 | 67.8 | 51.5 | **23.4** | 1.8 |
| English - Greek | 39.3 | 69.5 | 48.6 | 17.3 | 2.5 |
| English - Arabic | 43.9 | 71.3 | **53.5** | 16.4 | 2.7 |
| English - French | 47.7 | **71.4** | 52.5 | 18.9 | 2.9 |
| English - Spanish | 143.5 | 70.4 | 51.8 | 18.7 | **3.3** |
| Greek - English | 39.1 | 69.1 | 47.9 | 16.2 | 2.2 |
| Arabic - English | 43.3 | 69.3 | 50.7 | 22.5 | 2.3 |
| French - English | **50.5** | 71.2 | 52.7 | 19.7 | 2.3 |
| Spanish - English | 40.4 | 69.9 | 51.7 | 20.2 | 2.0 |

English model had the highest accuracy and so on. Therefore, table 2 shows that the results of this fine-tuning approach are not language dependent and our hypothesis that language is a confounder for our results is not true.

It is also important to note that this Marian-MT model has an embedding size of 512, which is twice less than the mBART model (and the LC model) we use in section 5.1. But because our goal in this section is to study the impact of languages and there are many pretrained models available of Marian-MT, we choose to use this model in our language study.[5]

### 5.3 How robust this fine-tuned model is with the distribution shift?

As also studied in Lample & Charton (2019), it is important to see whether these transformer models are biased towards the distribution of their training data or not. In order to evaluate this concept, we define two different kinds of distribution shift as follows:

- The first one is only for the integration task and is similar to section 4.7 described in Lample & Charton (2019). Meaning that we will investigate how robust our models trained in 5.1 are when we change their testing distribution. We report the evaluation metrics trained and tested on a different combination of training datasets in table 3.

- The second kind of distribution shift that we are interested in is due to the modality of the test dataset. This type of distribution shift was not studied by Lample & Charton (2019) and is a new type of distribution shifts we introduce in this paper. Each training sample we use on all tasks (in sections 5.1, and 5.2) has a combination of all different types of equations such as polynomial, trigonometric, and logarithmic expressions. We want to see whether a model trained on this type of dataset can generalize to solve type-dominant functions (i.e, functions containing only polynomial equations or containing only trigonometric equations and so on.). Therefore, we generate different types of test data, varying in the kind of equation they represent, such as trigonometric equations, polynomial equations, and logarithmic equations. We test the ability of our models trained in 5.1 to see which kinds of equations they can solve better, helping us to understand the impact of linguistic data better. The results are reported in table 4.

Table 3 indicates that our mBART model is more robust with respect to the generation distribution shift (i.e., FWD, BWD and IBP method for integration task.) and can achieve comparable performance in comparison to the pure transformer model (LC) model.

---

[5]Investigating the effect of embedding size more systematically to the results is considered as future work.

To evaluate the robustness of our approach in terms of different equation types, we created three different test datasets for each task. The first dataset is polynomial dominant, meaning that the samples of dataset were created mostly by polynomials without using trigonometric and logarithmic functions. The second and third datasets are trigonometric dominant and logarithmic dominant, respectively. This means that the trigonometric dominant dataset was created using mostly trigonometric functions, and the logarithmic dataset was generated using mostly logarithm and exponential functions. Table 4 indicates that our mBART model is not able to generalize to type dominant equations as well as the LC model can (except in the FWD and BWD approaches of the integration task.). The highest accuracies of both models are in their generalization to solve trigonometric expressions, and the lowest results are in pure polynomial ones. This agrees with our theory (see section 3), because the mBART model tries to find the shortest sequence and the higher order polynomial equations are less compressible. Also, higher order polynomials need accurate precision (F64) for their representation. On the other hand, trigonometric and the logarithmic equations can be compressed into shorter expressions (for example, $sin^2(x) + cos^2(x)$ is 1. or $e^{ix} = \cos x + i \sin x$), and ;therefore, the performance on these two sets of type-dominant test samples are better.

Table 3: Accuracy of the models (in percentage (%)) on function integration. Results are tested on test data sets of size 5000 samples. The models are trained on the 1 million sample size training data, as dicussed in section 5.1.

| | Forward | | Backward | | Integration by parts | |
|---|---|---|---|---|---|---|
| Training data | Ours(mBART) | LC | Ours(mBART) | LC | Ours(mBART) | LC |
| FWD | 87.38 | 79.42 | 7.30 | 6.90 | 74.20 | 74.10 |
| BWD | 12.82 | 9.28 | 92.24 | 83.40 | 24.02 | 17.60 |
| IBP | 30.46 | 28.70 | 35.00 | 20.50 | 86.26 | 87.44 |

Table 4: Accuracy of our models (in percentage (%)) on the integration and differential equation solving for different pretrained languages. Results are reported on test datasets of different types (polynomial, trigonometric and logarithmic.), and of size 5000.

| Testset Type | Metrics | Integration (FWD) | Integration (BWD) | Integration (IBP) | ODE (order 1) | ODE (order 2) |
|---|---|---|---|---|---|---|
| Polynomials | Ours | 60.6 | 67.8 | 70.7 | 39.1 | 8.9 |
| | LC | 54.7 | 60.0 | 80.1 | 60.6 | 57.9 |
| Trigonometric | Ours | 91.9 | 87.0 | 78.9 | 48.3 | 10.6 |
| | LC | 92.4 | 85.8 | 91.8 | 74.4 | 60.6 |
| Logarithmic | Ours | 90.9 | 75.1 | 72.4 | 35.9 | 6.8 |
| | LC | 87.9 | 73.3 | 87.96 | 75.6 | 72.0 |

# 6 Related work and Discussion

## 6.1 Transformers in different modalities

Attention (Bahdanau et al., 2014) is a powerful mechanism led to recent achievements in developing strong DNN models in NLP like the transformer architecture (Vaswani et al., 2017). Attention mechanism has also been used in other tasks such as visual explanation (Fukui et al., 2019), video captioning (Yan et al., 2019), healthcare (Choi et al., 2016), object detection (Chorowski et al., 2015), and speech recognition (Li et al., 2020). The transformer architecture introduced in (Vaswani et al., 2017) is an autoencoder that encodes the input data and then decodes them to the target domain. It does not use recurrent modulus and just uses self-attention mechanism. It is a breakthrough in NLP and is the base for many language models including bidirectional encoder representations from transformers, BERT, (Devlin et al., 2019), generative pretrained transformer, GPT-3, (Brown et al., 2020), Text-to-Text Transfer Transformer, T5, (Raffel et al., 2020) and Google's Meena (Adiwardana et al., 2020). It has also been successfully used as a baseline in other tasks such as object detection (Carion et al., 2020), image generation (Chen et al., 2021), image colorization (Kumar et al., 2021), video understanding (Sun et al., 2019), and visual question answering (Tan & Bansal, 2019). Furthermore, Yun et al. (2019) showed that transformers can universally approximate sequence to sequence functions. Therefore, the transformer is a

good choice for transfer learning not only because of their prosperity across different tasks, but also because of its architecture which makes it possible to use the hardware parallelism to train much more big models with much more training data.

## 6.2 SYMBOLIC COMPUTATION RELATED WORKS

The research on algebraic manipulation systems through computer is quite mature. The early work of solving symbolic integration were the heuristics programs written in LISP. They were named SIN (Symbolic INtegrator), SAINT, and SOlDIER (SOLUtion of Ordinary Differential Equations ROUTINE) (Moses, 1967). The obvious motivation during those programs, is the use of symbolic systems as an adjunct to numerical integration programs which involves parameters. SAINT program of symbolic integration shows the capability of a freshman calculus student. Thus, an unmodified SAINT pe.g.,am was of limited use in a practical algebraic system. More powerful programs follow, e.g., MATLAB project by MITRE Corporation, which solves integration of rational functions as good as sophomore college students. Though the capabilities of these programs are quite impressive, they mainly use tree search and matching algebraic expressions (pattern matching) as their workhorse. The program started showing its inherent limitation for those expressions which are not integrable in closed form, e.g., $\int e^{x^2} dx$ or $\int \frac{e^x}{x} dx$. Though there were some attempts of using Edge heuristics to solve those wild integrals, they were mainly unsuccessful. The era of deep neural networks ushers a new hope of solving the symbolic tasks by representing (encoding) the algebraic expressions in a feature space (Lample & Charton, 2019; Arabshahi et al., 2018; Allamanis et al., 2017; Zaremba et al., 2014; Loos et al., 2017; Trask et al., 2018; Łukasz Kaiser & Sutskever, 2016; Zaremba & Sutskever, 2015; Valipour et al., 2021; Ling et al., 2017; Polu & Sutskever, 2020). So instead of pattern matching on the raw mathematical expressions done in the pre-deep learning era programs, these deep models solve the algebraic systems in the feature space. These works on representing the symbolic expressions in a continuous and differential space using deep net architectures show the fundamental difference in the philosophy from the early SIN, SAINT, and SOlDIER pograms. The advantages of using deep net architectures are remarkable in terms of solving the algebraic systems approximately e.g., for those integrals which have no closed form solutions, and the average time complexity to solve. The deep models even started to show creativity on solving complex mathematical expressions, e.g., representing a mathematical expression in multiple ways. Very recently the research community started using language base transformer neural networks to solve symbolic computations (Lample & Charton, 2019; Hendrycks et al., 2021). The mathematical expressions are encoded as a sequence and a transformer is trained for a sequence-to-sequence translation task. The dot product attention module in the transformer architecture solves symbolic tasks efficiently. Saxton et al. (2019) takes a different route and created a large symbolic mathematics data set. All these research directions point towards the direction of solving mathematics is no more in the genre of human creativity, but a data problem. The unreasonable effectiveness of symbolic mathematics data and large neural architectures show the inevitable future of machine generated mathematical prover and symbolic mathematics.

## 7 CONCLUSION

Considering success of the transformer architecture in many tasks (Lu et al., 2021), including both language and symbolic mathematics, we proposed transfer learning from a pretrained language model with the transformer architecture for the downstream task of solving symbolic mathematical problems such as integration and differential equations. Using multiple experimental evaluation, we showed that these models could achieve competitive performance (specially in the integration tasks) with transformers fully trained on the symbolic math task without being pretrained on linguistic data. We showed that the language that the transformer model has been pretrained on does not have a significant impact in this transfer learning. We also evaluated that a model fine-tuned using our approach generalizes better in distribution shift scenarios for integration tasks.

REPRODUCIBILITY

We have provided all the code and data needed to re-produce our results as the supplementary materials. The Zip file contains a README.pdf file explaining how to run the experiments. The code for the LC model is run by the configurable scripts available on the Lample & Charton (2019)'s GitHub repository [6].

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
