# OpenReview forum: "Pretrained Language Models are Symbolic Mathematics Solvers too!"
_ICLR.cc/2022/Conference — ICLR 2022 Submitted_

### Official Review · Reviewer_4ER6 · 2021-10-28

**Correctness:** 3
**Technical Novelty And Significance:** 2
**Empirical Novelty And Significance:** 4
**Recommendation:** 6
**Confidence:** 5

**Main Review:**

This paper demonstrates that transformer pre-trained on NLP tasks can be used on non-linguistic, symbolic, problems. Given the amount of data needed to train large transformers, and the availability of models pre-trained on language problems, this is an important subject. There is some prior work on the subject, notably for theorem proving (https://arxiv.org/abs/2009.03393), but I know of no systematic study like this one. My main concern with the paper is the way the comparisons are handled. In particular, the models compared have very different sizes, which makes the effect of pre-training difficult to measure. Also, the difference in accuracy between the models complicates the sample size and out of distribution accuracy comparisons. These flaws in the experimental protocol weaken the authors' conclusions. Correcting them would greatly reinforce the paper.

Here is a series of observations and suggestions that might help improve the paper. The most important once (on experimental comparisons, are points 5-8):

1- page 2 : "trees are order invariant", this is not true : for instance, the left and right nodes of operators '-' and '/' cannot be exchanged. A more correct statement would be that the correspondance between mathematical expressions and the objects they represent is a many to one, but the same is true for natural language. Therefore, it is unlikely that pre-training on NLP tasks can eliminate this redundancy.

2- page 2 : the claim "Thus, our method is more robust than Lample & Charton (2019) as it is invariant to the data set generation method" seems incorrect: table 4 shows that whereas your model fares a little better than L&C when tested on a different generator than the one used at training, it is by no means invariant to the generation method (see also point 8).

3- page 3 : section 3, target language is "romanian" (instead of roman)?

4- page 4: Can you explain why the read path in figure 1 represents an expression of shortest length? The expression represented here is 3x^2 + e^x - (1-x^2), and is certainly not the shortest representation of this mathematical object: something like the tree representation of 4x^2 - 1 + exp(x) would probably be the shortest (and even 4x^2 + exp(x) if this is a solution to the integration problem, as you can set the constant to 0 to save tokens).

5- page 5-6 : Table 1 and figure 2: the mBart model has 12 layers and 16 attention heads, the LC model 6 layers and 8 attention heads. Therefore it is difficult to know whether the increase in accuracy over the FWD and BWD samples are due to the pre-training or to the larger model. Could the comparisons be done with a larger LC model (or a smaller mBart), to minimize the effect of model size?

6- You mention that mBart is biased toward shorter solutions. Can this bias be measured, for instance by comparing over a fixed test set the length of the solutions found by LC and mBart? Also, is it possible to measure if this bias has an impact on the accuracy of mBart? Could it be deactivated and the model tested without it? It is not clear whether this "short solution bias" can improve accuracy. In fact, it might even account for the bad performance over the more difficult ODE sets (by introducing a bias that complicates the model's work).

7- page 6: Your results suggest that mBart learns faster than LC (this is after all the goal of pre-training). However, the sample size comparison,1M for mBart vs 40-80M for LC, is not correct. 40-80M is the size of the data sets L&C created, not the number of samples needed to train their model to a certain accuracy level. Besides, the difference in size and accuracy between mBart and LC complicate the assessment. I think a correct metric would be to compare the number of samples needed by LC to achieve the same accuracies as mBart after 100k, 500k and 1M examples (judging from figure 2, this would probably suggest that mBart need half the samples LC use). This would not change your conclusion, but result in better supported claims.
As before, the comparison would be reinforced if the two models had the same number of layers and attention heads.

8- page 7: As it stands, table 3 does not really support your claim about the distribution shift. On the FWD trained set, mBart accuracy is 10% better than LC on the FWD test set, but only 5% on the BWD test set (and almost 0 on IBP).  For the BWD trained set, the conclusions go in the other way. Table 4, as you observe, is in favor of LC (especially if the higher accuracy of mBart is taken into account), but this may be due to the short solution bias (as you note). I believe a more precise analysis is necessary in order to back up your claim.

9- the GPT-f paper by Polu and Sutskever (https://arxiv.org/abs/2009.03393) is prior work, and should be in the related works. Together with language data (webcrawl in their case) they tried pretraining from Github code, and math articles from MathOverflow and Arxiv (their WebMath dataset) and report slight improvements in final accuracy. It might be interesting to experiment with such different pre-training tasks, to see whether "more topical" pre-training data (i.e. computer science and maths, instead of common text) helps.


**Summary Of The Paper:**

The authors use transformers pre-trained on machine translation tasks on the integration and differential equation datasets proposed by  Lample and Charton (Deep Learning for Symbolic Mathematics, ICLR 2020). They show that pre-training on "pure language" tasks help models, in terms of learning speed, accuracy, and, to a lesser extent, out of distribution generalization. Pre-training on different language pairs, on the other hand, has little impact on performance.

**Summary Of The Review:**

Overall, the paper is interesting and addresses an important subject. However, the results could be made more convincing by making mBart and LC more comparable. This includes :
- comparing models with similar number of parameters and features
- testing the impact of the bias towards shorter expressions
- better measurement of the number of samples needed to train in both cases
- a better analysis of the distribution shift comparison

Hence my note of 6, which I would raise if these issues can be addressed.

---

> ### Author Response · Authors · 2021-11-21
> **Author Response to Reviewer 4ER6 (1/2)**
>
> Many thanks for your detailed comments and feedback. We will address all of your comments on typos (thanks for the careful catch), and also your suggestions to improve the writing. The changes to our submitted version are highlighted by red color in the text.
>
> The following include our Proposed fix/changes (which will be done by the Nov. 22 deadline), as well as the discussion of the feedback:
>
> - We rewrite the  sentence “We see an inherent limitation in their work of encoding the mathematical expressions as trees: trees are order invariant.” to “The inherent limitation between the many-to-one map between mathematical expression and tree encoding is partially regularized by the pre-training with the language translation.”
>
> - We agree with the reviewer about comment 2, and re-write the sentence “Thus, our method is more robust than Lample & Charton (2019) as it is invariant to the data set generation method.” to “Our method also shows marginal robustness to different types of data set generation method as shown in table 4.”
>
> - We have updated our Figure 1 and re-written the theory through the lens of c-Approximate Nearest Neighbor Search (c-ANN). The shortest length search in the sequence space is modified to search in the parameter space of the transformer model. The parameter space of the transformer is represented as translation space (after pre-training from English to Romanian) and fine-tuning space (during the fine-tuning of the model with symbolic expressions). The generalization of our transformer is thus matching between translation space and fine-tuning space. The translation (English-Romanian) space is encoded as a tree structure, and the prediction by the transformer model with a test mathematical expression is a binary search.
>
> - Regarding your 5th comment, we used Lampe and Charton’s official implementation of their model, which is available here: (https://github.com/facebookresearch/SymbolicMathematics). And if you check their training script here: (https://github.com/facebookresearch/SymbolicMathematics#training), the number of layers and also the number of attention heads are configurable and as you mentioned, because the mBart model has 12 layers and 16 attention heads, we used this configurable script to train the LC model with the same number of layers and attention heads as the mBart model. We specified the number of layers and attention heads of the LC model in section 4.1 only based on their paper, but in practice (and in our experiments), we’ve trained them using the same number of layers and attention heads as our mBart model. We will specify this point in our next submission before the Nov. 22 deadline. In fact, as you mentioned, to have a fair and kind of apple-to-apple comparison, we controlled for every training parameter (such as learning rate, batch size, number of layers, …).
>
> - We view mBart is biased toward shorter solutions from the  principle of “Minimum Description Length Principle (MDL)”. More the mBart can compress the  sequence of mathematical expressions, the more it is able to find the regularities in them. This dependence on regularity and compressibility favors the shorter solutions by mBart.  The shorter bias can be measured as suggested by the reviewer by Synthetic Variations of mathematical expressions data set and testing both with LC and mBart.
>
> - Regarding your comment 7, we agree that changing the metric comparing the number of samples needed by LC to achieve the same accuracies as mBart after 100k, 500k, and 1M examples would help us express the results better. But the cost of training the models (Both LC and mBart) is very high, and such searching schema might take several days to accomplish the desired results. Therefore, we will not plan to do further experiments in this section.
>
> ** Continues in the following

---

> > ### Author Response · Authors · 2021-11-21
> > **Author Response to Reviewer 4ER6 (2/2)**
> >
> > - We agree with the reviewer, and we believe our model shows the marginal generation distribution shift because of the compression and regularity of sequences generated by different procedures.
> >
> >     1. In the **Forward generation (FWD)** method, the training input is generated as follows: a random function $f$ is generated, and its output is generated integral $\int f$ from the computer algebra system. The integral $\int f$ generated by the computer algebra system  is of regular structure. The computer algebra system discards any irregular $\int f$. Thus, the forced regularity introduced by the computer algebra system on $f$ allows for more compression by the  mBart model, and it gives better performance than the LC.
> >     2. In the Backward generation (BWD) method, the training input is a random function $f$ and the output is derivative $f'$ computed by mathematical softwares. $f'$ shows less regularity,  and almost any derivative of $f$ can be calculated fast by the softwares. The mathematical softwares does not require additional constraints while computing the derivatives. This less regularity of mathematical sequences allows less compression, and mBart shows marginal improvement.
> >     3. The Backward generation with integration by parts (IBP) method is a combination of  the Forward generation (FWD) and Backward generation (BWD). Hence, negligible regularity and compression, and the mBart model shows negligible improvement.
> >
> > - We agree with the reviewer and refer to the work GPT-f paper by Polu and Sutskever. We agree with the recommendation of pre-training the mBart model with WebMath dataset for interesting insight, but the philosophy of our paper is to show **natural** language helps solve the symbolic tasks. The WebMath dataset is more of a structured nature while human natural language is loose, less structured, and context-dependent. In our paper, we try to show how pre-training with this loose, less structured, and context-dependent language helps in solving the more structured and logical symbolic tasks. Though these two languages (human language and mathematical expressions) are apparently different, their representation inside the mBart model is similar. And mBart model while generalizing is actually solving an approximate nearest neighbor search.
> >
> > - Experiments regarding the **expression length** hypothesis as well as **pre-training on math texts** instead of regular English texts are considered as future work. We will try to do such exploration in the next versions of our paper.

---

> ### Author Response · Authors · 2021-12-08
> **A kind reminder**
>
> Dear Reviewer 4ER6,
>
> Thank you very much for all your efforts during the review. We still sincerely look forward to your reply, and we are wondering if our response has addressed your concerns. We appreciate your time and consideration.
>
> Many thanks,
>
>  Paper3389 Authors

---

### Official Review · Reviewer_3m2T · 2021-11-02

**Correctness:** 2
**Technical Novelty And Significance:** 1
**Empirical Novelty And Significance:** 2
**Recommendation:** 1
**Confidence:** 4

**Main Review:**

Strengths:
- This paper indeed seems to show you can sometimes get performance and efficiency improvements over a baseline transformer trained from scratch by using a pretrained transformer.

Weaknesses:
- This paper is unclear and poorly written. There were many sentences I didn't understand (to a very unusual extent).
- The evaluation setup seems strange to me. If they want to investigate the effect of language pretraining (of some sort) on symbolic mathematics, why focus exclusively on translation? For a number of experiments, they also focus exclusively on English to Romanian; while they assess other languages as well later on, they find that the specific pair of languages can make a big difference, so I'm not sure how much to trust the results with just English and Romanian.
- Presumably they could have easily used more standard pretrained language models (e.g. GPT-2 or T5) but don't do so for some reason. This would make the evaluation more compelling.
- The paper also makes a number of claims that seem unsupported or at least strange and confusing. For example, they say "Therefore, our mBART transformer model is lazy and during the fine-tuning method neural weights along the breadth are almost frozen and only the weights along criteria-satisfied path are updated. This method is also called stunting during its training" I don't know what this means. There were *many* other similarly confusing sentences.
- Their approach, which is *extremely* simple and just amounts to fine-tuning a pretrained model, also doesn't even consistently do better. For example, it seems to degrade performance on ODEs by quite a bit.

**Summary Of The Paper:**

This paper builds on Lample & Charton (2019) by assessing the ability of large pre-trained language models to solve symbolic mathematics problems. They show that transformers trained on language-to-language translation tasks can improve the efficiency final accuracy of learning symbolic mathematics in some settings.

**Summary Of The Review:**

This paper is poorly written and makes confusing + unsupported claims, the evaluation isn't thorough at all, and their approach doesn't even seem to help consistently very much anyway.

---

> ### Author Response · Authors · 2021-11-20
> **Author Response to Reviewer 3m2T**
>
> Thanks for your comments. We want to clarify the goal of our paper to address your comments about the experimental setup and the contributions of the paper, hoping that you will increase your evaluation of the paper.
>
> **Lample & Charton (2019)’s paper** provides a new insight to solve symbolic mathematics using a transformer-based model (encoder-decoder architecture) and why we can view this problem as a **translation** task using seq-2-seq models. This paper is very well-known as a baseline for deep learning for symbolic mathematics. Therefore, we approach the same schema and focus on the translation aspect of this idea, with a goal to shine some light on the capability of such models to transfer from language to math.
>
> The goal of our paper is to study whether we can improve symbolic math solving with the use of a pertained **transformer-based** model on pure language data. For this goal, we set up an empirical study by asking several research questions. In the first research question (section 5.1) we want to see whether the use of pre-trained models can help in using fewer data for training, which our results show that this hypothesis is true, specifically for the function integration tasks. Then, in section 5.2, we aim to see whether the language these models have been pre-trained on has any impact on the results or not. Which we see from table 2 of the paper that the results are language-independent, and that is the reason we use the English-Romanian pre-trained model in sections 5.1, and 5.3. In these two sections, we fine-tune our mBart model (as well as the LC model) on a dataset of size 1 million. Therefore, to reduce computational costs and by knowing that the results will be language-independent, we use the English-Romanian pre-trained model available online by Facebook (we refer to this model in the footnotes of section 5.1).
>
> We have two main reasons for choosing the mBart and the Marian-MT model:
> 1. The first reason is that these two models have several pre-trained versions available publicly for the Hugging-Face framework. This allows us to conduct our study without the need to pay considerable computation costs for training them.
> 2. The second reason is that these two transformer models are designed for the **language translation** tasks, and therefore, they are practical for our study. Moreover, if we want to have a fair comparison with the transformer-based model by Lample & Charton (2019), we require choosing a pre-trained encoder-decoder transformer model. That is why we chose mBart model, where both of the models (mBart and the model by Lample & Charton (2019)) have the same encoder-decoder architecture. Also, the goal of the paper is not to investigate the differences in the transformer models, and therefore, we think that the mBart and the Marian-MT models are sufficient to support our claims.
>
> Given the above discussion, we believe that our work is the first work in the literature that does an empirical study on the impact of pre-training language models on solving symbolic math tasks such as integration and differential equations. It is also worth stating that our experimental evaluation is trying to do an empirical study of the impact of the pertaining with language data to solve symbolic mathematical tasks, which we observed that helps significantly in the function integration task but fails for the differential equation solving. Our goal was not to propose a state-of-the-art model for these tasks.

---

> ### Author Response · Authors · 2021-12-08
> **A kind reminder**
>
> Dear Reviewer 3m2T,
>
> Thank you very much for all your efforts during the review. We still sincerely look forward to your reply, and we are wondering if our response has addressed your concerns. We appreciate your time and consideration.
>
> Many thanks,
>
>  Paper3389 Authors

---

### Official Review · Reviewer_2CuZ · 2021-11-07

**Correctness:** 2
**Technical Novelty And Significance:** 2
**Empirical Novelty And Significance:** 3
**Recommendation:** 3
**Confidence:** 4

**Main Review:**

Strength:

1. This work provided some empirical results on whether pretraining in machine translation could help with solving symbolic integration and solving differential equations.

2. The analysis on the choice of language pairs in pretraining and the generalization under distribution shift helps withe the understanding of the effect of pretraining.

Weakness:

1. The title "Pretrained Language Models are Symbolic Mathematics Solvers too!" is misleading. It suggests the pretrained language model is able to solve symbolic math problems reasonably well (without further training), but it actually can not until it is trained / fine tuned on a symbolic math dataset and the effect from pretraining is just faster training. Plus, it has already been shown in [Lu et al, 2021] that pretraining in language can help other modalities, so it is not so surprising that it can help in learning to solve symbolic math problem.

2. The presentation of results is confusing. The paper claims that "We achieve comparable accuracy on the integration task with our pretrained model while using around 1.5 orders of magnitude less number of training samples than the state-of-theart model in Lample & Charton (2019)", but it is unclear to me which result it is referring to. Both table 1 and table 3 states "The number of training samples used to train the models in all tasks is 1 million" and "the models are trained on the 1 million sample size training data". Figure 2 is explicitly shows the training dataset size and it doesn't seem like the mBART can outperform LC with 1 order of magnitude less data. Since this is one of the main claim of this paper, it would help to make the supporting evidence more clear and highlighted.

3. The result of "LC" in this paper seems much lower than the results presented in [Lample & Charleston, 2019], for example, Section 4.4 in their paper shows accuracy over 95% in FWD, BWD and IBP. What is the reason for the difference?

4. Section 3 ("Theory") is unclear. Some statements like "our model differs from the statistical machine learning theory, and uses a phenomena of self-regularization to find the optimum hypothesis Θ." are quite confusing. It might be helpful to use a concrete example or some equations to help explain the theory that the authors want to convey.


**Summary Of The Paper:**

This work investigate the problem of whether pretraining on language task such as machine translation could help with solving symbolic mathematics problems. Specifically they focused on solving symbolic integration and differential equations using the dataset similar to [Lample & Charleston, 2019]. The authors argued that finetuning a pretrained transformer model could get similar or better accuracy with fewer epochs. They then investigated the effect of the choice of language pairs and how the model works under domain shifts.


**Summary Of The Review:**

This paper provides some empirical study of whether pretraining in machine translation could help with solving symbolic integration and differential equations, and showed promising results. However, the effect that pretraining on language tasks could help other modalities has already been show in previous works so that novelty is limited. Furthermore the current draft contains several flaws: the title and some claims are not well supported; some of the writing, for example, Section 3, is confusing.

---

> ### Author Response · Authors · 2021-11-20
> **Author Response to Reviewer 2CuZ (1/2)**
>
> Many thanks for your comments. We want to clarify the goal of our experimental evaluation to address your comments 1-3 of yours (In the weakness part), hoping that you will increase your evaluation of the paper.
>
> In section 5.1 of the paper, to have an apple-to-apple comparison, we trained **“both” models** (i.e, the LC model and our mBart model) with one million data samples, and that is the reason the accuracies reported for the LC model is less than what they have reported in their paper and also in their GitHub repository (https://github.com/facebookresearch/SymbolicMathematics). As they report in their (like section 4.4 as you mentioned), those accuracies which are over 95% are achieved with sample sizes of 20-40 Million (please refer to section 4.1 of the Lample & Charleston paper). But here in table 1, table 3, table 4, and figure 2 of our paper, we trained the Lample & Charleston’s model (seq2seq) also with a 1 million data sample size (which we report in the captions.). The reason that we chose one million was due to our goal of alleviating the need for large training data. We aim to **have control over training size** and understand how important the training size is, and whether we can achieve comparable accuracy to the LC model while fixing the same configuration of parameters (The parameters that we controlled for include the training sample size, number of layers, number of attention heads, etc.). Note that the parameters like the number of layers and the number of attention heads are configurable in Lample & Charleston’s public code, and we set them to be the same as our mBart model (i.e, 12 layers and 8 attentions heads.). In other words, we wanted to have control over the capacity of the models to remove this potential confounding factor in our study. We will make the above statements clearer in our submitted version.
>
> Table 1 and figure 2 clearly show that in all the integration tasks (i.e, FWD, BWD, and IBP), this approach pertaining to language data helps our model to outperform the LC model while using the same number of training samples for both of them. This is the reason we claim **“We achieve comparable accuracy on the integration task with our pre-trained model while using around 1.5 orders of magnitude less number of training samples than the state-of-the-art model in Lample & Charton (2019)”.**
>
> It is also worth stating that our experimental evaluation is trying to do an empirical study of the impact of the pertaining with language data to solve symbolic mathematical tasks, which we observed that helps significantly in the function integration tasks but fails for the differential equation solving. In other words, we did not aim to propose a state-of-the-art model for these tasks. However, we laid out a comprehensive empirical study to investigate how much knowledge from pre-trained language models can be transferred to facilitate symbolic math tasks, and the key intuitions are described in the theory section. Specifically, solving the integration task is more similar to language translation (for instance, sin(x) and cos(x) convert to each other in derivation and integral.) and therefore, the pre-training phase significantly helped us improve the results. Whereas, we could not achieve similar good results on the ODE tasks, because they have much more levels of recursion and a more complicated task to convert to be similar to translation.

---

> > ### Author Response · Authors · 2021-11-20
> > **Author Response to Reviewer 2CuZ (2/2)**
> >
> > Regarding the concerns about the theory section, we will revise our text and highlight the changes (regarding clarifications) in **red** color. We will submit this version before the Nov. 22 deadline.
> >
> > **Proposed fix/changes (done):**
> >
> > We rewrite the sentence “Here our model differs from the statistical machine learning theory, and uses a phenomenon of self-regularization to find the optimum hypothesis Θ. Our Equation 1 of optimization explains part of the story to find the right hypothesis Θ. Big models like ours generalize from the phenomena of statistical physics and less from the optimization principle.” to:
> >
> > "Our model generalizes using the phenomena of self-regularization, which is a complex interpolation process between two parameter spaces. The parameter space $\Theta$ of our transformer model lie in a very high-dimensional space and is poorly understood. We model our parameter  space as a Normed space. The  learned parameter space $\Theta\_{translation} = (R^{d}, ||.||\_{\Theta\_{translation}})$ is a $d$-dimensional  normed space where $d$ is very large. The learned parameter space comes after the training from English to Romanian translation. Fine-tuning the model with mathematical expressions is a perturbation of the parameter space $\Theta\_{translation}$ to a new parameter space $\Theta\_{fine-tuning}$. The fine-tuning normed space can similarly be defined as $\Theta_{fine-tuning} = (R^{d}, ||.||_{\Theta\_{fine-tuning}})$. Our transformer model during the fine-tuning does a complex interpolation between normed spaces from $\Theta\_{translation}$ to $\Theta\_{fine-tuning}$ with a bounded variance. (Andoni, et al. "Hölder homeomorphisms and approximate nearest neighbors", Andoni, et al. "Data-dependent hashing via nonlinear spectral gaps")."
> >
> > We propose the generalization of our model through the  lens of c-Approximate Near Neighbor Search (c-ANN). During the fine-tuning of our model with a new test sample $Q \subseteq \Theta_{fine-tuning}$ of  symbolic expression, our model does an approximate nearest neighbor approximation search and find closet point in $P \subseteq \Theta_{translation}$ in the  translation metric space. For given $c \geq 1$ and $r > 0$ our transformer model finds point $P$ such that distance $d_{X}(q, p) \leq r$ with $c-$Approximate Near Neighbor Search (c-ANN). The  large size of the transformer parameter space, it is randomly partitioned into translation space $\Theta_{translation}$ and fine-tuning space $\Theta_{fine-tuning}$.  Each of the partition of translation space and fine-tuning space is prepossessed as a data structure of $Cell_{translation}$ and $Cell_{fine-tuning}$. The $Cell_{translation}$'s are then represented as a binary tree, as shown in our new added figure 1 (in the rebuttal version.). Given a query data structure  $Cell_{fine-tuning}$ during the symbolic mathematics tasks, our transformer model searches its $Cell_{fine-tuning}$ binary tree to find the approximate match. The normed space restrictions of $\Theta_{translation}$ and $\Theta_{fine-tuning}$ guarantees the  $c$-Approximate Near Neighbor Search (c-ANN) in polynomial time. The self-regularization process comes from the  underlying high-dimensional geometric structure of  $\Theta_{translation}$ and $\Theta_{fine-tuning}$ that makes it sparse.

---

> ### Author Response · Authors · 2021-12-08
> **A kind reminder**
>
> Dear Reviewer 2CuZ ,
>
> Thank you very much for all your efforts during the review. We still sincerely look forward to your reply, and we are wondering if our response has addressed your concerns. We appreciate your time and consideration.
>
> Many thanks,
>
>  Paper3389 Authors

---

### Official Review · Reviewer_oP2C · 2021-11-08

**Correctness:** 3
**Technical Novelty And Significance:** 1
**Empirical Novelty And Significance:** 1
**Recommendation:** 3
**Confidence:** 4

**Main Review:**

My main critique is that the paper omits closely related work - in particular GPT-f by Polu & Sutskever, which has already demonstrated gains on mathematical tasks after pretraining on natural language. I do not see that this paper adds a significant insight.

The claim stated in the title of the paper is not exactly true either. The pretrained models require a significant number of training steps in “fine-tuning” on datasets for symbolic mathematics to achieve a quality that is comparable to training them only on symbolic mathematics data.

The quality of the write-up is mixed. In particular the section called “Theory” is not ready to be published. For example:
- second line of section “Theory”: Roman -> Romanian?
- The next sentences read: “The learned parameter, Θ which is a matrix of neural weights, encodes the knowledge between any two sequences. Our model tries to predict the sequence of the shortest length by the principle of Occam’s razor.”
- The entire section is one long paragraph and I could not follow what it is saying. It appears to be a mix of intuition about what Transformer models are doing and a vague description of an approach?

Also in the introduction there are passages that need to be fixed:
- “and thus our model predict with better when the length …” -> “and thus our model’s predictions improve when the length …”
- “Section??” -> “Section 3”



**Summary Of The Paper:**

This paper improves studies if pretraining transformers on natural language improves their accuracy on symbolic mathematics tasks. The work follows the well-known paper by Lample & Charton and use the same distribution of (synthetic) data.

**Summary Of The Review:**

I am not convinced of the novelty of this work and the quality of the write-up is not up to ICLR standards.

---

> ### Author Response · Authors · 2021-11-20
> **Author Response to Reviewer oP2C (1/2)**
>
> Thanks for your comments, we will edit the paper to address the typos in our submitted version. We agree that we need to cite and discuss the paper GPT-f by Polu & Sutskever in our related-work section, but we want to clarify the difference between our paper with the GPT-f paper, hoping that you will increase your evaluation of the paper. There are some major and minor differences as follows:
>
> - Major:
>
>     The major difference between our work and the work of the GPT-f is the **task** we are attempting to accomplish. While GPT-f paper aims to do **Theorem Proving**, we use the established schema of converting the Symbolic Mathematics task to a **Language Translation** task, which has been initially investigated by Lample & Charton. Different from both GPT-f and Lample & Charton work, we conducted a comprehensive **empirical study** to investigate the impact of knowledge transfer from pre-trained language models to symbolic math tasks. In particular, in section 5, we investigated 3 transfer learning questions regarding (i) sample efficiency, (ii) language specificity of models, and (iii) robustness to distributional shift.
>
>     Note that this data can include something **like poetry**, a completely different data from the arXiv math data (and other datasets) that the GPT-f paper uses to train on. For example, in section 5.2, we study whether the pertained language has an impact on the results of our fine-tuning.
>
> - Minor:
>
>     1. The optimization scheme for mathematical proving and translation tasks is very different. In our paper, we are using the advantage of having a good weight initialization by pre-training the model on language data.
>     1. The two tasks that we study in our paper (function integration and ODE solving) are not studied in the GPT-f paper. This is important because, from the results of our experiments, we found out that solving the integration task is more similar to language translation (for instance, sin(x) and cos(x) convert to each other in derivation and integral.) and therefore, the pretraining phase significantly helped us improve the results. Whereas, we could not achieve similar results on the ODE tasks, because they have much more levels of recursion and a more complicated task to convert to be similar to translation. We elaborated on these in the theory section.
>
> Given the above discussion, we believe that our work is the first work in the literature that does an empirical study and shines some light on the impact of pretraining language models on solving symbolic math tasks such as integration and differential equations. However, we envision further works by investigating even more diverse symbolic math tasks such as linear algebra studied in another [paper]((https://openreview.net/pdf?id=L2a_bcarHcF)) submitted to ICLR .

---

> > ### Author Response · Authors · 2021-11-20
> > **Author Response to Reviewer oP2C (2/2)**
> >
> > Regarding our Theory section, we will apply the following modifications in our submitted version (Before the Nov. 22 deadline.)
> >
> > **Proposed fix/changes (done):**
> >
> > - We rewrite the sentence: “The learned parameter, $\Theta$ which is a matrix of neural weights, encodes the knowledge between any two sequences. Our model tries to predict the sequence of the shortest length by the principle of Occam’s razor.” to “The model parameters $\Theta$ as expressed in Equation 1 are learned with  gradient descent during the translation task.”
> > - We rewrite the sentence: “Our model shows heavy-tailed distribution property and thus our model predict with better when the length of the input mathematical expression and predicted output mathematical expression differ, as by the Backward generation (BWD) method Lample & Charton (2019). Changed to  “Our model’s predictions improve even when there is a difference of length between input and output sequence.  This is because of the phenomena of heavy-tailed distribution, where the model can generate rare small or large output expressions.”

---

> ### Author Response · Authors · 2021-12-08
> **A kind reminder**
>
> Dear Reviewer oP2C,
>
> Thank you very much for all your efforts during the review. We still sincerely look forward to your reply, and we are wondering if our response has addressed your concerns. We appreciate your time and consideration.
>
> Many thanks,
>
>  Paper3389 Authors

---

### Author Response · Authors · 2021-11-23
**General response to reviewers**

Dear reviewers,

Many thanks for your comments. We have updated a rebuttal version of our paper and highlighted the changes to the previous version in **red** color.
The following are the changes we have done and also some common clarifications.

- We believe that our work is the first work in the literature that does an **empirical** study on the impact of pre-training language models on solving symbolic math tasks such as integration and differential equations. It is also worth stating that our experimental evaluation is trying to do an empirical study of the impact of the pertaining with language data to solve symbolic mathematical tasks, which we observed that helps significantly in the **function integration** task but fails for the differential equation solving. **Our goal was not to propose a state-of-the-art model for these tasks.**
- We have updated our **Figure 1 and re-written the theory section** through the lens of c-Approximate Nearest Neighbor Search (c-ANN). The shortest length search in the sequence space is modified to search in the parameter space of the transformer model. The parameter space of the transformer is represented as translation space (after pre-training from English to Romanian) and fine-tuning space (during the fine-tuning of the model with symbolic expressions). The generalization of our transformer is thus matching between translation space and fine-tuning space. The translation (English-Romanian) space is encoded as a tree structure, and the prediction by the transformer model with a test mathematical expression is a binary search.
- We agree that we need to cite and discuss the paper GPT-f by Polu & Sutskever in our related-work section (which we added), but we want to clarify the difference between our paper with the GPT-f paper. The major difference between our work and the work of the GPT-f is the **task** we are attempting to accomplish. While GPT-f paper aims to do **Theorem Proving**, we use the established schema of converting the Symbolic Mathematics task to a **Language Translation** task, which has been initially investigated by Lample & Charton. Different from both GPT-f and Lample & Charton work, we conducted a comprehensive **empirical study** to investigate the impact of knowledge transfer from pre-trained language models to symbolic math tasks. In particular, in section 5, we investigated 3 transfer learning questions regarding (i) sample efficiency, (ii) language specificity of models, and (iii) robustness to distributional shift. Note that this data can include something **like poetry**, a completely different data from the arXiv math data (and other datasets) that the GPT-f paper uses to train on. For example, in section 5.2, we study whether the pertained language has an impact on the results of our fine-tuning.
- In the comparisons we have done in sections 5.1 and 5.3, every parameter of the LC model and the mBart model (such as the number of layers and the number of attention heads, …) **are the same** (please refer to section 4.1 in the paper.). The accuracies of the LC models reported in our paper are less than the ones reported in their main paper (https://arxiv.org/pdf/1912.01412.pdf), because in our paper we **train the LC model** with around **1.5 order of magnitude less data which is 1 million** samples, but the high accuracies in their paper are achieved with more data (20 - 40 million) samples as they state in section 4.1 of their paper (https://arxiv.org/pdf/1912.01412.pdf).

---

### Decision · Program_Chairs · 2022-01-20

**Decision:**

Reject

**Comment:**

This papers presents a method for solving symbolic mathematic tasks. It first pretrains a transformer model with language translation, and then fine-tunes the pretrained model to the downstream mathematic tasks. It contains interesting points but our reviewers have serious concerns which are not fully resolved in the rebuttal. For the integration task, the proposed method achieves good results comparing with Lample & Charleston 2019 (LC) with much less training data. However, as the authors also noted (see the rebuttal), the higher accuracies in LC are achieved with more data. If the authors could also at least show how much data the proposed method needs to achieve the best result in LC, it will be very helpful for understanding the value of this work. In addition, the proposed method did not show similar improvements on the ODE task. So it is hard to see how this proposed method can be generally useful. Our reviewers also have big concerns on writing. Many sentences are really confusing.